# Clinical Assessment of Neuroinflammatory Markers and Antioxidants in Neonates with Hyperbilirubinemia and Their Association with Acute Bilirubin Encephalopathy

**DOI:** 10.3390/children9040559

**Published:** 2022-04-14

**Authors:** Dina Abdel Razek Midan, Wael A. Bahbah, Noha Rabie Bayomy, Noha M. Ashour

**Affiliations:** 1Department of Pediatrics, Faculty of Medicine, Menoufia University, Menoufia 32511, Egypt; wael_bahbah@yahoo.com (W.A.B.); nohaashour800@gmail.com (N.M.A.); 2Department of Medical Biochemistry and Molecular Biology, Faculty of Medicine, Menoufia University, Menoufia 32511, Egypt; noharabie@yahoo.com

**Keywords:** neuroinflammation, antioxidants, hyperbilirubinemia, ABE, neonates

## Abstract

Objective: To assess the oxidant and antioxidant status in neonates with and without hyperbilirubinemia and their association with early manifestations of acute bilirubin encephalopathy (ABE), in addition to eliciting the possible oxidative effects of phototherapy. Methods: This prospective observational study was conducted with 104 full-term newborns at Menoufia University Hospitals from January 2020 to January 2021 to help resolve the debate regarding whether bilirubin is an antioxidant. The cases group (Group I) included 52 full-term newborns (37–40 weeks) with hyperbilirubinemia during the neonatal period, while the control group (Group II) included 52 healthy, full-term age and sex-matched newborns who did not have hyperbilirubinemia. The cases group was further subdivided into Group Ia (*n* = 12), which included newborns who presented with neurological manifestations suggesting early ABE, and Group Ib (*n* = 40), which included newborns with no signs suggestive of ABE. All newborns were subjected to clinical and neurological examinations, as well as laboratory investigations. Results: Comparing the specific biological markers between the Group 1 subgroups before phototherapy, the mean plasma levels of prostaglandin-E_m_, prostaglandin E_2_, and TSB were significantly higher in Subgroup I(a) (all *p* < 0.05). After phototherapy, Subgroup I(a) patients had significantly higher levels of prostaglandin-E_m_, DSB, and TSB (*p* < 0.05). The biological marker levels improved after phototherapy in terms of TAC (0.811 vs. 0.903), MDA (8.18 vs. 5.13), prostaglandin-E_m_ (37.47 vs. 27.23), prostaglandin E_2_ (81.09 vs. 31.49), DSB (1.21 vs. 0.55), and TSB (16.63 vs. 8.26; *p*-value < 0.05). Conclusion: Our study showed that an elevated level of serum bilirubin increases oxidative stress and decreases antioxidant capacity. The reduction in bilirubin levels by phototherapy is associated with a decrease in oxidative stress markers (MDA, PGE_m_, and PGE_2_) and an upsurge in TAC, highlighting the absence of oxidative stress effects arising from phototherapy. Neonates with neurological manifestations suggesting ABE had higher levels of oxidative stress markers and lower levels of total antioxidant capacity than those without.

## 1. Introduction

Hyperbilirubinemia is a disorder that occurs when total serum bilirubin exceeds 5 mg/dL, causing a yellowish discoloration of the skin, mucous membranes, and sclera [1]. About 60 to 80% of term and preterm neonates experience hyperbilirubinemia during the first week of life, and it is prolonged in 10% of breastfed neonates up to one month of age [2]. Hyperbilirubinemia is the most common clinical problem in the newborn period, and there are concerns that high levels of bilirubin may be potentially toxic to the central nervous system and lead to bilirubin encephalopathy [3,4,5]. 

Acute bilirubin encephalopathy is an infrequent neurological damage disorder that remains a critical problem globally [6,7,8]. It is induced by high levels of indirect bilirubin, which is fat-soluble, crossing the blood-brain barrier. This can be due to either increased production of indirect bilirubin (due to hemolysis, birth injuries, or polycythemia) or decreased excretion (due to hypoalbuminemia, hereditary bilirubin conjugation defects, and disruption or obstruction in the biliary system) [7,9,10,11,12,13,14]. The clinical spectrum of ABE ranges from mild symptoms to very severe manifestations; children may present with movement disorders, isolated hearing loss, and/or auditory dysfunction [4,15]. Oxidative stress is defined as a state in which oxidation exceeds the capacity of the antioxidant system in the body. Bilirubin has been identified as an antioxidant at low levels, as it scavenges reactive oxygen and nitrogen with high efficiency, while it becomes toxic at high concentrations [16,17].

Total antioxidant capacity (TAC) is a measure of the amount of free radicals scavenged by a test solution, and can be used to evaluate the antioxidant capacity of biological samples [18]. TAC, superoxide dismutase (SOD), and catalase levels are higher in neonates with hyperbilirubinemia without signs of ABE, and are lower in neonates with ABE. Consequently, bilirubin is proposed to have an antioxidant effect that is limited in ABE [19]. Malondialdehyde (MDA) is a secondary product produced during lipid peroxidation. It is considered a biomarker for oxidative stress [17,20]. Malondialdehyde is positively associated with hyperbilirubinemia [21]. 

Prostaglandins are lipid mediators synthesized by cyclooxygenase enzymes from arachidonic acid in response to specific stimuli, such as trauma or inflammation [22]. Prostaglandin-E2 (PGE_2_) is one of the most common types of prostaglandins, and has both inflammatory and anti-inflammatory effects [23,24]. It has been proposed that bilirubin indirectly stimulates its production, as it inhibited nitric oxide synthase and nitric oxide in rats given endotoxins [23]. Further, PGE_2_ is upsurged in the cerebrospinal fluid of neonates with acute hypoxia, and is associated with the degree of birth asphyxia [25].

PGE_2_ is unstable and is degraded into various metabolites, including prostaglandin-Em (PGEm). PGE_2_ and PGEm are rapidly elevated during an infectious event and may cause cardiorespiratory disturbances, which are the most common presenting symptoms of neonatal sepsis and autonomic dysfunction [26].

There is a debate about whether bilirubin is an antioxidant, or whether it sensitizes and activates the oxidative stress cascade [27]. Because of this disagreement, we intended to assess the lipid peroxidation and subsequent oxidative stress and antioxidant status in neonates with and without hyperbilirubinemia, and their association with early manifestations of acute ABE. Further, we aimed to measure the effect of phototherapy on these markers.

## 2. Materials and Methods

We conducted this prospective observational study with full-term newborns admitted to the Neonatal Intensive Care Unit of Menoufia University Hospitals from January 2020 to January 2021. The study protocol was approved by the Ethics Committee of the Faculty of Medicine, Menoufia University (IRB: 4-2018PEDI32) on 7 July 2019. The study was conducted in accordance with the Helsinki Declaration of 1964, as revised in 2013. We obtained written consent from the guardian of each neonate included in the study.

### 2.1. Patient and Public Statement

Public healthcare clinics, but not families, were involved in the content and design.

### 2.2. Eligibility Criteria

The study included full-term newborns (37–40 weeks) with indirect hyperbilirubinemia requiring phototherapy during the neonatal period.

Neonates meeting any of the following criteria were excluded from our study:Preterm neonates (aged < 37 weeks)Neonates with intrauterine growth retardation (IUGR)Neonates with congenital malformations or suspected inborn errors of metabolism or features suggestive of chromosomal disorderPresence of asphyxia, hypoxic ischemic encephalopathy, or sepsisNeonates with intracranial hemorrhage or respiratory distressNeonates with conjugated bilirubin >20% of total serum bilirubin or who needed an exchange blood transfusion as a line of managementNeonates whose parents/guardians refused to give an informed consent.

### 2.3. Study Groups

The included newborns were classified into two groups. 

Group I included newborns with indirect hyperbilirubinemia requiring phototherapy based on the American Academy of Pediatrics (AAP) guidelines [28]. Group I was further subdivided into Group I(a), which included newborns who presented with neurological manifestations suggesting early ABE, e.g., hypotonia, lethargy, high pitched cry and poor suckling, and Group I(b), which included newborns with no signs of ABE.Group II included healthy full-term age and sex-matched newborns who did not have hyperbilirubinemia, recruited from Menoufia University Hospital outpatient clinic during their routine follow-up visits.

### 2.4. Study Process and Evaluations

On admission, all newborns had a full history taken, including antenatal, natal, and postnatal histories, with an emphasis on risk factors for hyperbilirubinemia, along with a clinical exam including a detailed neurological examination where general appearance, mental status, cranial nerves, motricity, reflexes and coordination were assessed and a sensory exam was carried out [29]. Laboratory investigations, including (a) complete blood count (CBC) with differential reticulocyte count, using a Swelab Alfa Plus-Blood cell counter with a blood film, (b) total and direct serum bilirubin and renal function tests using a spectrophotometry kit supplied by Diamond Diagnostics Inc. (Holliston, MA, USA), (c) liver function tests using a spectrophotometry kit supplied by Human, (d) C-reactive protein using a spectrophotometry kit supplied by Biosystem, (e) blood culture if needed, using the DL D2mini microbial I.D. and AST system, Zhubi DLBiotec Co. Ltd. (Zhuhai, China), to exclude infection, and (f) serum blood glucose level and serum ammonia, were evaluated in patients with suspected ABE. These investigations were performed in the central laboratory of Menoufia Medical Hospital, following the manufacturer’s instructions. 

#### Radiological Examination for the Study Groups


Chest X-ray if needed.Brain MRI for suspected cases of ABE just before discharge using a 1.5 T scanner.


### 2.5. Blood Samples for the Specific Markers

Laboratory work was carried out in the Medical Biochemistry and Molecular Biology Department, Faculty of Medicine, Menoufia University. We collected 5 mL peripheral blood samples in two sterile tubes from all newborns included in the study, following complete aseptic precautions. The first contained ethylene diamine tetraacetic acid (EDTA) as an anticoagulant for plasma separation using 2 mL of the blood for prostaglandin E2 (PGE_2_) and PGE_2_ metabolites (PGE_M_) measurement. The other 3 mL of blood was left to clot for 30 min in plain serum tubes, and then centrifuged for 10 min at 4000 r.p.m. for serum separation for malondialdehyde (MDA) and total antioxidant capacity (TAC) measurements. Plasma and serum samples were immediately frozen at −80 °C until analysis. Plasma PGE_2_ and PGEm concentrations were measured in both the cases and control groups using commercially available ELISA kits supplied by Sun Red Bio-Technology Co., Ltd. (Shanghai-China), according to manufacturers’ instructions. 

In Group I, blood samples were obtained twice. The first sample was obtained just before starting phototherapy, and the second after phototherapy to measure PGE_2_, PGEm, MDA, and TAC levels. The type and duration of phototherapy were recorded. In Group II, we measured PGE_2_, PGEm, MDA, and total TAC levels once, at the time of presentation.

### 2.6. Statistical Analysis

All analyses were performed using the Statistical Package of Social Science (SPSS) version 21 (SPSS Inc., Chicago, IL, USA). We presented the qualitative data in frequencies and percentages; the chi-squared test (χ^2^) was used to assess the association between two or more variables, while the Fischer’s exact test was used if the expected cell count of more than one-quarter of cases was less than five. Continuous data are presented as the mean ± standard deviation (SD) or median and interquartile range (IQR) for parametric and non-parametric data. To compare quantitative variables between two groups, the Student t-test and Mann-Whitney U test were used for parametric and non-parametric data, respectively. The paired t-test was used to assess the statistical significance of the difference between two population means of dependent (paired) samples. Wilcoxon signed-rank tests were used to compare two quantitative variables in two dependent (paired) groups when the data were non-normally distributed. Receiver operator characteristics (ROC) analysis was performed to assess the biomarker performance, with respective maximum accuracy points for sensitivity and specificity. A significant difference was considered present when the *p*-value was less than 0.05. 

## 3. Results

### 3.1. Characteristics of the Included Population

A total of 104 full-term newborns were included in our study, and were divided into two groups. Group I included 52 newborns (32 male and 20 female) with indirect hyperbilirubinemia requiring phototherapy, while Group II included 52 age- and sex-matched clinically healthy newborns who did not have hyperbilirubinemia. Group I was further classified into Group I(a), that included 12 newborns with neurological manifestations suggesting early ABE, while Group I(b) included 40 newborns with no signs of ABE. 

We detected no statistically significant differences between the groups regarding the baseline characteristics (*p* > 0.05). Moreover, both groups had comparable body temperatures, respiratory rates, and weights (*p* > 0.05), while Group I patients had a slightly higher mean blood pressure (50.03 ± 3.29 versus 48.85 ± 2.77, *p* = 0.046), as shown in Table 1.

Comparing the demographic data among the two subgroups of Group I, an increased frequency of vaginal delivery and intensive phototherapy among Subgroup I(a) compared to Subgroup I(b) participants (66.7% vs. 20.0%; *p* = 0.004) and (66.7% vs. 35%; *p* = 0.034) was found. The mean duration of phototherapy was significantly higher in Subgroup I(a) compared to Subgroup I(b) patients (114.5 vs. 98.3; *p* = 0.032). No significant differences regarding gender, maternal age, gestational age, and neonatal age were observed (*p* > 0.05). Subgroup I(a) and Subgroup I(b) showed comparable body temperatures, respiratory rates, mean blood pressures, and weights (*p* > 0.05), Table 2.

### 3.2. Biological Markers

At the time of admission (before phototherapy), we observed significantly higher levels of TSB, DSB, MDA, and PGEm, as well as a lower level of TAC, among Group I patients compared to Group II participants (*p* < 0.05), Table 3. We detected no significant difference between the groups regarding plasma levels of PGE_2_ (*p* = 0.059). 

As demonstrated in Table 4, compared to the biological markers before phototherapy, the levels were improved after phototherapy in terms of TAC (0.811 vs. 0.903), MDA (8.18 vs. 5.13), PGEm (37.47 vs. 27.23), PGE_2_ (81.09 vs. 31.49), DSB (1.21 vs. 0.55), and TSB (16.63 vs. 8.26); *p*-value < 0.05). Regarding the specific biological markers in the Group 1 subgroups, before phototherapy, the mean plasma levels of PGE_M_, PGE_2_, and TSB were significantly higher among Subgroup I(a) than Subgroup I(b) (54.39 vs. 32.40, 59.25 vs. 35.63, and 21.83 vs. 15.7; *p*-values= 0.019, 0.019, and 0.006). After phototherapy, Subgroup I(a) patients had significantly higher levels of PGEm, DSB, and TSB (0.021, 0.001, and 0.001), Table 5.

### 3.3. Efficacy of the Studied Biological Markers as Diagnostic Biomarkers for ABE

The usefulness of TAC, MDA, PGEm, PGE_2_, and MDA/TAO ratio as diagnostic markers of ABE was tested through receiver-operating characteristic (ROC) curve analysis, as shown in Table 6 and Figure 1. At 33.34 points, the PGE_M_ showed a sensitivity and specificity of 66.7% and 70%, with an area under the curve of 72.5%. Moreover, at 43.37 points, PGE_2_ showed a sensitivity and specificity of 66.7% and 85%, with an area under the curve of 72.5%.

### 3.4. Biological Markers and Different MRI Findings

The levels of MDA, PGEm, and PGE_2_ were all higher in cases with abnormal MRI findings than in those with normal MRIs, unlike TAC, which was lower but all did not reach a level of significance (Table 7). 

## 4. Discussion

Previous studies have shown contradictory results for the association between bilirubin and oxidative stress and the antioxidant system. The primary mechanisms of bilirubin-induced neuronal damage have not clearly been identified and have different pathways, including bilirubin-induced lipid peroxidation, neuroinflammation, and excitotoxicity, in addition to sustained energy failure [27]. The current study highlighted the variations in serum levels of a group of neuroinflammatory markers for lipid peroxidation and TAC in neonates with hyperbilirubinemia and its complications, as represented by ABE, as well as the effect of phototherapy on these neuroinflammatory markers.

Our study showed that neonates with hyperbilirubinemia had significantly high mean levels of MDA and PGEm and low levels of TAC compared to the control group (8.18 vs. 1.98, 37.47 vs. 20.74, and 0.811 vs. 1.09, *p* < 0.05, respectively). These results matched those of Yigit et al. [21], who suggested that hemolytic hyperbilirubinemia may cause oxidative stress by raising levels of MDA (9.1 vs. 5.2, *p* < 0.05). The increased levels of MDA and PGEm in Group I can be explained by neuroinflammation induced by oxidative stress, resulting in overexpression of COX-2 in neurons that mediate the production of MDA and PGE_2_ [30]. Our results revealed that TAC was significantly lower in the cases compared to the control group, which is consistent with results reported by Dani et al. [31] (0.74 vs. 0.99, *p* < 0.05). In contrast to our TAC results, Gopinathan et al. [32] revealed that antioxidant capacity was significantly raised with bilirubin in term neonates, implying its antioxidant activity. Bilirubin may be associated with oxidative stress as its production is accompanied by the production of iron and carbon monoxide due to heme oxygenase (HO) activity; this could be explained by the physiological degradation of heme into biliverdin, which is further converted to bilirubin, iron, and carbon monoxide by HO [27,33]. On the other hand, the decrease in bilirubin may be associated with the reduction in oxidative stress in neonates [31]. In the current research, the rise of PGEm and PGE_2_, which are indicators for lipid peroxidation in neonates with hyperbilirubinemia and ABE, supports our finding that bilirubin is an oxidizing agent [16,21]. 

Phototherapy is the first line of treatment for neonatal hyperbilirubinemia. It may lead to oxidative stress, especially in neonates with limited antioxidant protective capacity [27,34]. Many studies have tested the possible oxidative effects of phototherapy, such as that of Ayyappan et al. [35], who measured MDA as a marker for lipid peroxidation and glutathione peroxidase, reduced glutathione, and catalase antioxidants in neonates with jaundice before and after phototherapy. They showed that MDA and glutathione peroxidase levels after phototherapy were higher than before phototherapy (7.22 vs. 5.72 and 86.33 vs. 79.23, *p* < 0.05, respectively). In addition, they demonstrated that antioxidant enzymes of glutathione and catalase were reduced after phototherapy (16.7 vs. 12.4 and 8.91 vs. 7.23, *p* < 0.05). Another study in favor of the oxidative effects of phototherapy was carried out by Iskander et al. [34], who revealed decreased TAC and increased total oxidative stress after eight hours of phototherapy. Shahab et al. [19] also demonstrated that MDA was increased with decreased total antioxidant capacity and reduced glutathione after phototherapy. This could be explained by the previous assumption that bilirubin may be a photosensitizer that absorbs light energy, then transferring it to oxygen, leading to the formation of reactive oxygen which may increase oxidative stress [36]. As a sensitizer, bilirubin associated with RBC surfaces may stimulate lipid peroxidation of the plasma membrane of the RBCs, leading to its hemolysis and an increase in oxidative stress [36,37]. In contrast to the abovementioned studies, our research reported that lipid peroxidation parameters (MDA, PGE_m_, and PGE_2_) after phototherapy were lower than before phototherapy (5.13 vs. 8.18, 27.23 vs. 37.47, and 31.49 vs. 81.09, *p* < 005, respectively). Moreover, TAC was increased after phototherapy (0.811 vs. 0.903, *p* < 0.05), indicating that phototherapy has no oxidative stress effects. Other studies have also been in line with our results regarding effects of phototherapy on neuroinflammatory markers and TAC, revealing that phototherapy is safe and effective in treating hyperbilirubinemia [36,38,39]. The favorable effects of phototherapy on the oxidant–antioxidant status in our work could be explained by the idea that bilirubin is toxic at high concentrations, and when it is broken down by phototherapy its neuroinflammation-induced production of cytokines and prostaglandin decreases, and its toxic effects fade away [16,17].

The findings of our research proposed no significant difference regarding TAC and MDA between cases with and without signs of ABE. On the other hand, PGE_m_ and PGE_2_ were significantly higher in neonates with ABE, which significantly decreased after phototherapy (*p* < 0.05). 

By studying the different inflammatory biomarkers and their ability to differentiate cases with signs suggestive of acute ABE from those without, our study suggested that PGE_2_ had the best discriminatory power and could be considered a good negative marker, as 85% of jaundiced neonates were correctly diagnosed as not having ABE (specificity). On the other hand, PGEm is a stable and reliable biomarker for identifying jaundiced newborns at risk of ABE (sensitivity 67%). On interpreting the MRI results of cases, there were non-significant differences between cases with abnormal MRI findings and those with normal results and the biological marker levels, and this could have been affected by the small sample size of the subgroups, as well as by the early identification of cases, which reduced the likelihood for abnormal MRI findings to appear. Many previous studies have stated that the earliest MRI changes tend to show a high T-1 signal in the affected regions, which also can be seen as part of normal brain development, such as during myelination. Further, patients with kernicterus may not necessarily show any abnormalities on an MRI. A 2008 case series by Gkoltsiou et al. [40]. reported the inexplicable conclusion that, while all children with severe cerebral palsy and a history of hyperbilirubinemia had abnormal central grey matter on later scans, the characteristic central grey matter MRI features of kernicterus were not seen in early scans. The current research has some limitations: more lipid oxidation and antioxidant parameters need to be measured in bigger samples and multicenters, with prolonged follow-up of neonates, to address any adverse outcomes. Serial MRI scans should be done to obtain better correlations.

## 5. Conclusions

In conclusion, the current study revealed that high bilirubin levels are associated with increased oxidative stress and decreased antioxidant capacity. In contrast to many other studies, our study revealed that the reduction in bilirubin levels achieved by phototherapy is associated with a decrease in oxidative stress markers (MDA, PGEm, and PGE2) and an upsurge in TAC. Additionally, PGEm and PGE_2_ have moderate diagnostic efficacy for the detection of ABE. 

## Figures and Tables

**Figure 1 children-09-00559-f001:**
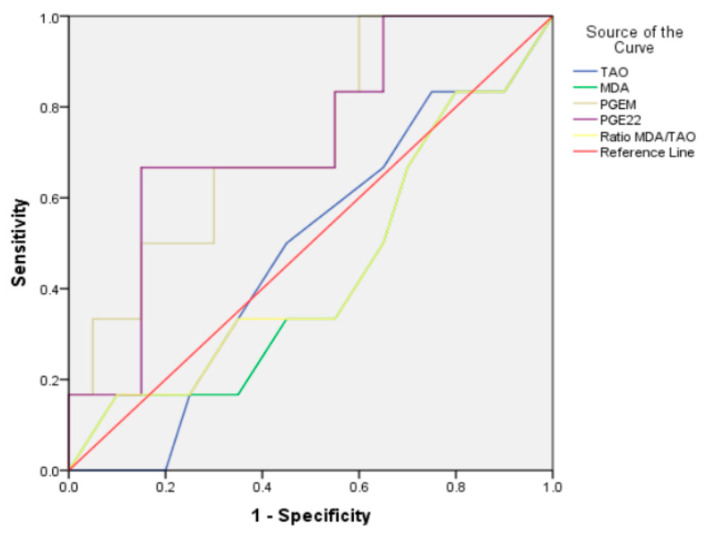
Diagnostic performance of biological markers for ABE in Group I patients.

**Table 1 children-09-00559-t001:** The demographic and clinical characteristics of the studied groups.

	Group I (N = 52)	Group II (N = 52)	*p* Value
Demographic Data			
Maternal age (years), Mean ± SD	27.19 ± 4.22	28.53 ± 4.05	0.097 (NS)
Gender, *n* (%)			
Male	32 (61.5)	30 (57.7)	0.689 (NS)
Female	20 (38.5)	22 (42.3)	
Gestational age (weeks), Mean ± SD	37.962 ± 0.766	38.08 ± 0.737	0.435 (NS)
Neonatal age (days) Mean ± SD Median (range)	3.42 ± 2.099 3 (1.00–11.00)	3.27 ± 1.33 3.5 (1.00–7.00)	0.633 (NS)
Consanguinity, *n* (%)			
Yes	22 (42.3)	24 (46.2)	0.780 (NS)
No	30 (57.7)	28 (53.8)	
Mode of delivery, *n* (%)			
Cesarean	36 (69.2)	40 (76.9)	0.377 (NS)
Normal Vaginal	16 (30.8)	12 (23.1)	
**Clinical Characteristics**			
Temperature (°C), Mean ± SD Median (IQR)	37.1.7 ± 0.20 37.1 (36.78–37.23)	36.93 ± 0.23 37 (36.8–37.1)	0.079 (NS)
Respiratory rate (cycle/minute), Mean ± SD	48.19 ± 4.09	47.92 ± 2.09	0.783 (NS)
Mean Blood pressure (mm/Hg), Mean ± SD	50.03 ± 3.29	48.85 ± 2.77	0.046 (S)
Weight (grams), Mean ± SD	3084.23 ± 480.03	3207.69 ± 386.41	0.140 (NS)
Albumin	3.05 ± 0.182	3.117 ± 0.167	0.094 (NS)
Apgar score	9.02 ± 0.828	9.115 ± 0.832	0.556 (NS)

S = Statistically significant (*p* < 0.05), NS = Non-significant.

**Table 2 children-09-00559-t002:** The demographic and clinical characteristics of Subgroup I(a) and Subgroup I(b).

	Group I (a) (N = 12)	Group I (b) (N = 40)	*p* Value
Demographic Data			
Maternal age (years), Mean ± SD	26.0 ± 4.26	27.55 ± 4.14	0.264 (NS)
Gender, *n* (%)			
Male	6 (50.0)	26 (65.0)	0.500 (NS)
Female	6 (50.0)	14 (35.0)	
Gestational age (weeks) Mean ± SD	38.17 ± 0.937	37.90 ± 0.709	0.307 (NS)
Neonatal age (days) Mean ± SD Median (IQR)	2.5 ± 0.798 3 (2.00–3.00)	3.70 ± 2.289 3 (2.00–5.00)	0.10 (NS)
Consanguinity, *n* (%)			
Yes	6 (50)	16 (40.0)	0.665 (NS)
No	6 (50)	24 (60.0)	
Mode of delivery, *n* (%)			
Cesarean	4 (33.3)	32 (80.0)	0.004 (S)
Normal Vaginal	8 (66.7)	8 (20.0)	
Type of phototherapy, *n* (%)			
Conventional	4 (33.3)	26 (65)	0.034 (S)
Intensive	8 (66.7)	14 (35)	
**Duration of Phototherapy (Hours)**			
Mean ± SD Median (range)	114.5 ± 24.37 118 (70.0–141.0)	98.3 ± 61.2 70 (45.0–260.0)	0.032 (S)
**Clinical Characteristics**			
Temperature (°C) Mean ± SD Median (IQR)	36.95 ± 0.34 36.9 (36.6–37.2)	36.85 ± 0.70 37.1 (36.8–37.275)	0.220 (NS)
Respiratory rate (cycle/minute), Mean ± SD	48.50 ± 4.62	48.1 ± 3.92	0.767 (NS)
Mean blood pressure (mm/Hg), Mean ± SD	49.17 ± 2.79	50.3 ± 3.38	0.295 (NS)
Weight (grams), Mean ± SD	3229.17 ± 107.04	3040.75 ± 532.72	0.232 (NS)
Albumin	3.065 ± 0.179	3.017 ± 0.195	0.425 (NS)
Apgar score	9.125 ± 0.822	8.667 ± 0.779	0.094 (NS)

Group I (a): a subgroup of group I presented with ABE, Group I (b): a subgroup of group I with no ABE, S = Statistically significant (*p* < 0.05), NS = Non-significant.

**Table 3 children-09-00559-t003:** Biological markers of the studied groups (before phototherapy).

	Group I (N = 52)	Group II (N = 52)	*p* Value
Total antioxidant capacity (TAC) (mMl/L)	0.811 ± 0.09	1.09 ± 0.25	<0.001 (S)
Malondialdehyde (MDA) (Nmol/mL)	8.18 ± 1.86	1.98 ± 0.37	<0.001 (S)
Prostaglandin Em, (PGEm) (ng/L)	37.47 ± 23.63	20.74 ± 5.42	0.002 (S)
Prostaglandin E2 (PGE2) (ng/L)	81.09 ± 23.92	31.62 ± 5.35	0.059 (NS)
Direct serum bilirubin (DSB) (mg/dL)	1.21 ± 0.59	0.25 ± 0.15	<0.001 (S)
Total serum bilirubin (TSB) (mg/dL)	16.63 ± 3.73	2.29 ± 2.21	<0.001 (S)

All values are presented as Mean± SD, S = Statistically significant (*p* < 0.05), NS = Non-significant.

**Table 4 children-09-00559-t004:** Biological markers in Group I before and after receiving phototherapy.

	Before Phototherapy (N = 52)	After Phototherapy (N = 52)	*p* Value
Total antioxidant capacity (TAC) (mMl/L)			
Mean± SD	0.811 ± 0.09	0.903 ± 0.49	<0.001 (S)
Median (IQR)	0.8 (0.77–0.88)	0.92 (0.85–0.93)	
Malondialdehyde (MDA) (Nmol/mL)			
Mean ± SD	8.18 ± 1.86	5.13 ± 1.27	<0.001 (S)
Median (IQR)	8.25 (6.20–9.75)	5 (4.35–5.9)	
Prostaglandin Em, (PGEm) (ng/L)			
Mean ± SD	37.47 ± 23.63	27.23 ± 6.87	<0.001 (S)
Median (IQR)	29.81 (25.47–38.47)	26.75 (23.03–32.32)	
Prostaglandin E2 (PGE2) (ng/L)			
Mean ± SD	81.09 ± 23.92	31.49 ± 6.09	<0.001 (S)
Median (IQR)	37.34 (928.19–45.26)	32.01 (27.31–36.36)	
Mean Direct serum bilirubin (DSB) (mg/dL)			
Mean ± SD	1.21 ± 0.59	0.55 ± 0.41	<0.001 (S)
Median (IQR)	1 (0.84–1.57)	0.5 (0.30–0.53)	
Total serum bilirubin (TSB) (mg/dL)			
Mean ± SD	16.63 ± 3.73	8.26 ± 2.17	<0.001 (S)
Median (IQR)	15 (13.8–18.7)	8.75 (6.15–10.5)	

S = Statistically significant (*p* < 0.05).

**Table 5 children-09-00559-t005:** Biological markers in Group 1 subgroups before and after phototherapy.

		Before Phototherapy			After Phototherapy		
		Group I (a) (N = 12)	Group I (b) (N = 40)	*p* Value	Group I(a) (N = 12)	Group I(b) (N = 40)	*p* Value
Total antioxidant capacity (TAC) (mMl/L)	Mean ± SD	0.803 ± 0.09	0.814 ± 0.09	0.827 (NS)	0.890 ± 0.04	0.907 ± 0.05	0.188 (NS)
	Median (Range)	0.82 (0.63–0.89)	0.8 (0.63–0.96)		0.91 (0.83–0.93)	0.92 (0.83–0.98)	
Malondialdehyde (MDA) (Nmol/mL)	Mean ± SD	7.80 ± 1.87	8.295 ± 1.84	0.458 (NS)	5.25 ± 1.35	5.10 ± 1.25	0.512 (NS)
	Median (Range)	7.45 (5.6–10.9)	8.9 (5.6–10.9)		5.25 (3.0–7.30)	5 (3.0–7.30)	
Prostaglandin Em, (PGEm) (ng/L)	Mean ± SD	54.39 ± 36.99	32.40 ± 14.67	0.019 (S)	31.18 ± 8.90	26.05 ± 5.66	0.021 (S)
	Median (Range)	38.27 (29.04–63.79)	29.495 (22.36–36.41)		33.03 (16.10–42.691)	25.55 (15.99–37.31)	
Prostaglandin E2 (PGE2) (ng/L)	Mean ± SD	59.25 ± 42.41	35.63 ± 9.71	0.019 (S)	32.84 ± 3.65	31.1 ± 6.57	0.247 (NS)
	Median (Range)	46.28 (28.06–128.91)	36.64 (15.37–73.31)		32.56 (27.54–37.77)	31.86 (19.38–45.72)	
Mean Direct serum bilirubin (DSB) (mg/dL)	Mean ± SD	1.31 ± 0.655	1.18 ± 0.57	0.359 (NS)	0.667 ± 0.20	0.52 ± 0.44	0.001 (S)
	Median (Range)	1.25 (0.3–2.1)	1 (0.5–2.8)		0.6 (0.5–1.0)	4 (0.1–2.0)	
Total serum bilirubin (TSB) (mg/dL)	Mean ± SD	21.83 ± 2.72	15.7 ± 2.31	0.006 (S)	10.07 ± 0.66	7.73 ± 2.15	0.001 (S)
	Median (Range)	22 (17–25)	14.5 (12–20)		10.05 (9.30–11.0)	7.75 (4.5–11.30)	

S = Statistically significant (*p* < 0.05), NS = Non-significant.

**Table 6 children-09-00559-t006:** Diagnostic performance of biological markers for ABE in Group I patients.

	Best Cut Off Point	Area under the Curve	Sensitivity	Specificity
Total antioxidant capacity (TAC)	0.815	0.521	50%	45%
Malondialdehyde (MDA)	7.45	0.429	50%	35%
Prostaglandin Em (PGEm)	33.345	0.725	66.70%	70%
Prostaglandin E2 (PGE2)	43.37	0.725	66.70%	85%
MDA/TAO ratio	9.587	0.554	66.7%	55%

TAO = total anti-oxidant.

**Table 7 children-09-00559-t007:** Association between MRI and Biological markers in kernicterus subgroup.

	Abnormal MRI (N = 7)	Normal MRI (N = 5)	Test of Significance	*p* Value
Total antioxidant capacity (TAC)	0.83 ± 0.04	0.7 ± 0.13	t = 1.5	0.194 (NS)
Malondialdehyde (MDA)	7.8 ± 2.2	8.1 ± 1.6	t = −0.2	0.8 (NS)
Prostaglandin Em (PGEm)	47.1 ± 14.6	57.9 ± 18.9	t = −0.4531	0.66 (NS)
Prostaglandin E2 (PGE2)	53.3 ± 16.2	61.2 ± 22.3	t = −0.2857	0.78 (NS)
Total serum bilirubin (TSB)	17.4 ± 2.4	15.9 ± 7.2	t = 0.4703	0.65 (NS)
Direct serum blirubin (DSB)	1.1 ± 0.6	1.5 ± 0.96	t = −0.70	0.5 (NS)

(NS) Nonsignificant, (MDA) Malondialdehyde, (PGE_2_) Prostaglandin E_2_, (PGEm) Prostaglandin Em, (DSB) Direct serum bilirubin, (TSB) Total serum bilirubin, Abnormal MRI findings e.g., (1) T1 weighted wave: Increased signal intensity in bilateral Globus pallidus interna and Globus pallidus externa, mesencephalon, caudate nucleus and lentiform nucleus. (2) T2 weighted wave: Increased signal intensity in hippocampus, minimally decreased signal intensity due to whit matter myelination delay.

## Data Availability

The data presented in this study are available on request from the corresponding author.

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
