# Peer review of "Clinical Assessment of Neuroinflammatory Markers and Antioxidants in Neonates with Hyperbilirubinemia and Their Association with Acute Bilirubin Encephalopathy"

_children, 2022, doi:10.3390/children9040559_

Round 1
Reviewer 1 Report
This is an important paper as we attempt to understand the balance in anti-oxidant and oxidative stress for bilirubin and the most widespread treatment phototherapy. Thanks for doing this important study.
Your message would be much clearer and its significance to those treating hyperbilirubinemia much clearer with shorter well written sentences clearly stating which results are from your study and which are from the literature.
Some rewording is needed but over all it reads well and your points are clear i.e. Line 14 might be better if worded debate about whether bilirubin is an antioxidant or not.
Lines 28-30 The reduction of bilirubin levels by phototherapy is associated with a decrease in oxidative stress markers........
Lines 40 and 41 antioxidant misspelled.
Line 41 needs to be reworded for clarity.
Line 81-84. There is a debate about whether bilirubin is an antioxidant or whether it sensitizes......
Line 120 On admission all newborns had a full history taken along with a clinical exam including a neurological examination,....
Line 216 Previous studies showed contradictory results about......
Line 227-228 need to be reworded for clarity
Lines 235-238 also confusing and need to be reworded for clarity
Lines 239-240 Phototherapy is the first line of treatment of neonatal hyperbilirubinemia. It may lead........
Lines 261-264. not clear at all what you are trying to say---please reword---also most would not agree that a bilirubin of more than 12mg/dl is toxic in a term infant. Most would use a much higher number.
Line 267-269 Again unclear what you are trying to say and how much is your work and how much is others.
Also several run on sentences including lines 272-275
Line 278 -280. The reduction of bilirubin levels seen in phototherapy is associated with a decrease in oxidative stress markers............ In contrast to many other studies, in our study, this reduction of bilirubin levels with phototherapy was also associated with upsurges in the TAC. Additionally, PGEm and PGE2......
A few of your terms should be updated to terms more commonly used now i.e. Kernicterus is now generally reserved for beyond the neonate and acute bilirubin encephalopathy is the usual term for changes seen in the neonatal period.
Author Response
Reviewer 1:
Comment: This is an important paper as we attempt to understand the balance in antioxidant and oxidative stress for bilirubin and the most widespread treatment phototherapy. Thanks for doing this important study.
Response: Thank you very much for your appraisal and your constructive comments.
Comment: Your message would be much clearer and its significance to those treating hyperbilirubinemia much clearer with shorter well written sentences clearly stating which results are from your study and which are from the literature.
Response: Thank you for your comment. We modified the confusing points and edited the paper's language by editing service belonging to MDPI.
Comment: Some rewording is needed but over all it reads well and your points are clear i.e. Line 14 might be better if worded debate about whether bilirubin is an antioxidant or not.
Response: Thank you for pointing this to us. As requested, we modified line 14 and edited the paper's language by editing service belonging to MDPI.
Comment: Lines 28-30 The reduction of bilirubin levels by phototherapy is associated with a decrease in oxidative stress markers........
Response: We modified it.
Comment: Lines 40 and 41 antioxidant misspelled.
Response: We corrected it and and edited the paper's language by editing service belonging to MDPI.
Comment: Line 41 needs to be reworded for clarity.
Response: We corrected it and edited the paper's language by editing service belonging to MDPI.
Comment: Line 81-84. There is a debate about whether bilirubin is an antioxidant or whether it sensitizes......
Response: We edited it and edited the paper's language by editing service belonging to MDPI.
Comment: Line 120 On admission all newborns had a full history taken along with a clinical exam including a neurological examination,....
Response: We corrected it and edited the paper's language by editing service belonging to MDPI.
Comment: Line 216 Previous studies showed contradictory results about......
Response: We corrected it and and edited the paper's language by editing service belonging to MDPI.
Comment: Line 227-228 need to be reworded for clarity
Response: We corrected it and edited the paper's language by editing service belonging to MDPI.
Comment: Lines 235-238 also confusing and need to be reworded for clarity
Response: We corrected it and edited the paper's language by editing service belonging to MDPI.
Comment: Lines 239-240 Phototherapy is the first line of treatment of neonatal hyperbilirubinemia. It may lead........
Response: We corrected it and edited the paper's language by editing service belonging to MDPI.
Comment: Lines 261-264. not clear at all what you are trying to say---please reword---also most would not agree that a bilirubin of more than 12mg/dl is toxic in a term infant. Most would use a much higher number.
Response: We corrected it and edited the paper's language by editing service belonging to MDPI.
Comment: Line 267-269 Again unclear what you are trying to say and how much is your work and how much is others.
Response: We corrected it and edited the paper's language by editing service belonging to MDPI.
Comment: Also several run on sentences including lines 272-275
Response: We modified them and edited the paper's language by editing service belonging to MDPI.
Comment: Line 278 -280. The reduction of bilirubin levels seen in phototherapy is associated with a decrease in oxidative stress markers............ In contrast to many other studies, in our study, this reduction of bilirubin levels with phototherapy was also associated with upsurges in the TAC. Additionally, PGEm and PGE2......
Response: We modified them and edited the paper's language by editing service belonging to MDPI.
Comment: A few of your terms should be updated to terms more commonly used now i.e. Kernicterus is now generally reserved for beyond the neonate and acute bilirubin encephalopathy is the usual term for changes seen in the neonatal period.
Response: We changed them. Thank you.
Reviewer 2 Report
This study examined the association between neuroinflammatory markers in neonates with hyperbilirubinemia. The study findings showed increased oxidative stress and decreased antioxidant capacity in neonates with hyperbilirubinemia. However, the research methods are not enough to provide sound evidence.
I have some suggestions for the authors and hope that my comments are constructive.
- In this study, suspected kernicterus symptoms alone should not be used to be considered as the possibility of kernicterus. (definition in this study: newborns presented with neurological manifestations suggesting early kernicterus). This is a major study flaw.
- This study used Group Ia to compare with Ib. The results indicated a dose-response impact in the biomarkers and presented oxidative effects at different disease severities. However, neonates in Group Ia were suspected cases of kernicterus, and there are doubts about whether it conforms to a causal relationship. Please consider testing the effect to see if there is a dose-response impact by the bilirubin levels.
- For the different oxidation and antioxidant tests in the manuscript, it is suggested to add a comparison table between previous literature and this study to understand the essential information quickly.
- Please add information about the beds for treating hyperbilirubinemia or other neonatal diseases in your institution.
- English revision is required. There are typos in the manuscript.
- Please clearly provide the information at what age the Control group was tested.
Author Response
Reviewer 2:
Comment: This study examined the association between neuroinflammatory markers in neonates with hyperbilirubinemia. The study findings showed increased oxidative stress and decreased antioxidant capacity in neonates with hyperbilirubinemia. However, the research methods are not enough to provide sound evidence. I have some suggestions for the authors and hope that my comments are constructive.
Response: Thank you very much for your appraisal and your constructive comments and we will take all your comments in consideration.
Comment: In this study, suspected kernicterus symptoms alone should not be used to be considered as the possibility of kernicterus. (definition in this study: newborns presented with neurological manifestations suggesting early kernicterus). This is a major study flaw.
Response: We aimed in our study to early pick the abnormalities in neurological examination of the studied neonates and correlate this with the neuroinflammatory markers changes; hence it could be used plus bilirubin levels to early diagnose neonates at risk of kernicterus not to diagnose the full-blown picture of it, especially in developing countries were MRI studies are not available in every center. Many previous studies stated that the earliest MRI changes tend to show a high T-1 signal in the affected regions, which also can be seen as part of normal brain development such as myelination,and patients with kernicterus may also not necessarily show any abnormalities in MRI.A 2008 case series by Gkoltsiou et al. reported the inexplicable conclusion that all children with severe cerebral palsy and a history of hyperbilirubinemia had abnormal central grey matter on later scans, the characteristic central grey matter MRI features of kernicterus were not seen in early scans. In our study, all cases already had done MRI imaging but not in the early stage; instead, it was done at discharge to give enough time, so if there are any abnormalities, it could be seen in the scans. We have done an association between these MRI findings and the tested biological markers in cases but a non-significant difference was elicited, so we preferred not to include extra non -significant data in our article, especially as it is not our direct aim. We attached the table of association in our revised manuscript (table 7).
Comment: This study used Group Ia to compare with Ib. The results indicated a dose-response impact in the biomarkers and presented oxidative effects at different disease severities. However, neonates in Group Ia were suspected cases of kernicterus, and there are doubts about whether it conforms to a causal relationship. Please consider testing the effect to see if there is a dose-response impact by the bilirubin levels.
Response: We apologize for not understanding which dose-response and testing effects you mean, so it would be kind if you could give us an extra explanation and clarification about this point.
Comment: For the different oxidation and antioxidant tests in the manuscript, it is suggested to add a comparison table between previous literature and this study to understand the essential information quickly.
Response: Thank for your comment. We compared our results with the results of previous literature in the discussion section. Our study is a primary research study not literature review or a meta-analysis study. We will follow your instructions in our following studies.
Comment: Please add information about the beds for treating hyperbilirubinemia or other neonatal diseases in your institution.
Response: Our institution have a total number of thirty incubator, of them, five incubators are specified for treating nenoates with hyperbilirunemia.
Comment: English revision is required. There are typos in the manuscript.
Response: We edited the paper's language by editing service belonging to MDPI.
Comment: Please clearly provide the information at what age the Control group was tested.
Response: We mentioned it in table 1.
Reviewer 3 Report
I think this is a very interesting paper today, but there are a few problems.
Regarding exclusion criteria, patients with HIE, IVH, and other CNS diseases should be excluded.
About nuclear jaundice
What is the basis for the diagnosis of nuclear jaundice?
Were MRI and ABR performed?
How was motor function assessed?
Unbound bilirubin(UB) is bilirubin where the indirect bilirubin is not able to bind to albumin.
UB crosses the blood-brain barrier and is deposited in the basal ganglia and cerebellum, causing nuclear jaundice.
Did you measure your UB?
Did you measure the serum albumin level?
If possible, it should be listed in Table 1 and Table 2.
Table 1, 2
Why is there a significant difference in vaginal delivery?
It is better to describe Apgar Score.
Table 6
What is the AUC of the MDA/TAC ratio?  Isn't it supposed to be useful?
Author Response
Reviewer 3:
Comment: I think this is a very interesting paper today, but there are a few problems.
Response: Thank you very much for such an evaluation and your constructive comments.
Comment: Regarding exclusion criteria, patients with HIE, IVH, and other CNS diseases should be excluded.
Response: We already excluded them in the study, we mentioned this point in the method section.
Comment: About nuclear jaundice
What is the basis for the diagnosis of nuclear jaundice?
Response: Nuclear jaundice was initially diagnosed in our study by the presence of a history of risk factors, neurological examination, bilirubin levels requiring intervention, laboratory investigations to rule out other exclusion criteria which may be misinterpreted with nuclear jaundice, and finally, MRI studies which were done before discharge. We aimed in our study to early pick the abnormalities in a neurological examination of the studied neonates and correlate these with the neuroinflammatory markers changes hence it could be used plus bilirubin levels to early diagnose neonates at risk of kernicterus not to diagnose the full-blown picture of it, especially in developing countries were MRI studies are not available in every center.Many previous studies stated that the earliest MRI changes tend to show a high T-1 signal in the affected regions, which also can be seen as part of normal brain development such as myelination, and patients with kernicterus may also not necessarily show any abnormalities in MRI.A 2008 case series by Gkoltsiou et al reported the inexplicable conclusion that, while all children with severe cerebral palsy and a history of hyperbilirubinemia had abnormal central grey matter on later scans,the characteristic central grey matter MRI features of kernicterus were not seen in early scans. In our study, all cases already had done MRI imaging but not in the early stage instead, it was done at discharge to give enough time so if there are any abnormalities it could be seen in the scans. We have done an association between these MRI findings and the tested biological markers in cases but non-significant difference was elicited, so we preferred not to include extra non -significant data in our article, especially as it is not our direct aim. We will attach that table of association as a supplementary one.
Comment: Were MRI and ABR performed?
Response: All cases with signs suggesting acute bilirubin encephalopathy had done MRI studies just before discharge.
No ABR was performed as it is not available at our center and we only depended on clinical examination to elicit any auditory abnormalities.Comment: How was motor function assessed?
Response: We assessed it through newborn neurological assessment according to Volpe, J. 2004.
Comment: Unbound bilirubin(UB) is bilirubin where the indirect bilirubin is not able to bind to albumin.
UB crosses the blood-brain barrier and is deposited in the basal ganglia and cerebellum, causing nuclear jaundice.
Did you measure your UB?
Response: unbound bilirubin wasn’t measured as it is unavailable at our center.
Comment: Did you measure the serum albumin level?
If possible, it should be listed in Table 1 and Table 2.
Response: Yes, we measured it, and we added it to tables 1 and 2. Thank you for letting us know.
Comment: Table 1, 2
Why is there a significant difference in vaginal delivery?
It is better to describe Apgar Score.
Response: Our results revealed a significant difference between the subgroups regarding vaginal delivery, which wasn’t elicited in the main groups, but this is what we found.
It may be explained by that oxytocin is used in many vaginal deliveries at our center, and it is known from previous literature and studies that oxytocin is an antidiuretic hormone that causes hyposmolarity with subsequent swelling of red blood cells so, making them fragile and susceptible to hemolysis.
Apgar score values will be added to Tables 1 and 2.
Comment: Table 6
What is the AUC of the MDA/TAC ratio?  Isn't it supposed to be useful?
Response: MDA/TAC have low discrimination power to detect neonates with kernicterus, (AUC=0.55); consequently, we added it to the figure and the table 6.
Round 2
Reviewer 3 Report
I have no further comment.